# How well do simulated populations with GPT-4 align with real ones in clinical trials? The case of the EPQR-A personality test

Gregorio Ferreira
IN3, Universitat Oberta de Catalunya
Barcelona, Spain
jferreirade@uoc.edu

Jacopo Amidei
IN3, Universitat Oberta de Catalunya
Barcelona, Spain
jamidei@uoc.edu

Rubén Nieto
eHealth Research Lab, Universitat Oberta de Catalunya
Barcelona, Spain
rnietol@uoc.edu

Andreas Kaltenbrunner
IN3, Universitat Oberta de Catalunya
Barcelona, Spain
ISI Foundation
Torino, Italy
kaltenbrunner@gmail.com

## ABSTRACT

In this paper, we test if GPT-4o can simulate populations for clinical trials. We performed two experiments, with the Eysenck Personality Questionnaire-Revised (EPQR-A) in three different languages (Spanish, English, and Slovak). Our results show that GPT-4o displays specific personality traits which may vary depending on different parameter settings and questionnaire language. Furthermore, the question of whether simulated populations (mimicking real ones) can be created and used for testing questionnaires is still inconclusive. While we find encouraging results in some personality traits and differences between genders and study fields, we also observe that results for the virtual population answering the questionnaire differ from the ones found in real populations. Accordingly, further research is needed to test how to reduce the differences between virtual and real populations.

## KEYWORDS

LLM, GPT, Eysenck Personality Questionnaire-Revised (EPQR-A)

**ACM Reference Format:**
Gregorio Ferreira, Jacopo Amidei, Rubén Nieto, and Andreas Kaltenbrunner. 2024. How well do simulated populations with GPT-4 align with real ones in clinical trials? The case of the EPQR-A personality test. In *Proceedings of Artificial Intelligence and Data Science for Healthcare (AIDSH-KDD'24).* ACM, New York, NY, USA, 7 pages.

## 1 INTRODUCTION

In this paper, we explore to what extent Large Language Models (LLMs) can mimic human characteristics and if they can help researchers and/or healthcare professionals simulate populations for testing surveys or questionnaires. Questionnaires and surveys are efficient research methods for acquiring information about individuals and are beneficial for unearthing information that is not directly measurable.

Defining sound surveys and questionnaires is not reduced to providing a sequence of questions. The overall structure, flow, coherence, and adequacy of the questions have to be taken into account [41]. The two most important and time-consuming steps when defining a questionnaire are the pre-test (or pilot test), and the test of psychometric properties (for example, reliability and validity). Both steps are complex, since they involve the computation of multiple indices and applying the questionnaire to different populations. LLMs could simulate such populations for testing surveys or questionnaires, facilitating these two steps and reducing time consumption.

To advance on that, we studied to what extent GPT displays a specific personality pattern and can simulate populations that could be used to assess questionnaires. In particular, we used the EPQR-A questionnaire [12] that was designed to measure Extraversion (degree of being outward, and socially engaged), Neuroticism (degree of emotional stability), and Psychoticism (degree of impulsiveness, difficulty in accepting and following rules). This questionnaire is particularly compelling for our analyses because it is widely used in psychology and has been translated into several languages, allowing us to perform multi-language experiments.

In detail, our study aims to: 1) Test the baseline personality of GPT by using different experimental presentations and language versions (English-EN, Spanish-SP, and Slovak-SK) of the questionnaire. 2) Test the properties of the questionnaire in a simulated population sample – mimicking the characteristics of the real population from [15] and, 3) Check if specific groups of virtual personas (i.e. type of studies and genders) match the expected personality for them (taking into account available literature).

## 2 RELATED WORK

The recent surge in employing psychology methodologies within the LLMs framework (see for example [20, 22, 27]) has led [19] to coin the term *machine psychology*. Similarly, [37] introduces the term *AI Psychometrics*. According to Hagendorff, machine psychology seeks to uncover emergent abilities in LLMs that traditional natural language processing benchmarks cannot detect. Thus, the term aims to encompass various approaches that utilize psychological methods to analyze LLMs' behavior under one umbrella.

Among the various directions that contribute to the development of machine psychology, a popular one is studying the personality of LLMs. For example, the EPQR-A were used by [13] whereas the Big Five factors [9] were used, among others, by [26, 38, 37, 33] to quantify the personality traits of LLMs. Similarly, IPIP-NEO [17]

---
*AIDSH-KDD'24, August 26, 2024, Barcelona, Spain*
2024.

was used in [38], the Myers-Briggs Type Indicator (MBTI) [4] test was used in [28], and Short Dark Tetrad (SD4) [36] was used in [37]. In a slightly different fashion, [18] investigates LLM´s behavioral profile in a *dynamic* context instead of a *static* one. While the outcomes of the aforementioned studies may vary depending on the LLMs and questionnaires used, there is enough support to draw the promising and optimistic conclusion that personality assessments for LLMs are valid and reliable. These findings hold significance, considering that personality tests are tailored for humans, and there is no guarantee beforehand that they will yield valid and reliable results for LLMs.

A further endeavour, pursued by researchers such as [26, 24, 38], involves exploring the potential for adjusting LLMs' personalities. The objective is to study whether tailored prompts can push LLMs to replicate human personality traits. Encouragingly, findings from these papers suggest that LLMs can indeed be molded to imitate particular personality profiles.

A more general direction has brought researchers to utilize LLMs to create simulated humans, serving as experimental participants, survey respondents, or other agents. Simulated populations can streamline experiments, reduce time and costs, and can be utilized in studies unsuitable for human involvement. This concept, referred to by various names such as *guinea pigbots* [23], *silicon samples*, or *homo silicus* [21], has found application across various domains within social science. For example, the employment of LLMs as substitutes for human participants was studied in psychological research [23, 10, 35], political polling [39], software engineering research [16], teaching research [32], economics [21], social media platforms design [34, 44], market research to understand consumer preferences [3] and more generally social science research [2]. These studies yielded mixed results. While some outcomes closely mirrored the behaviors observed in real human counterparts, other research raised questions about the suitability of replacing human participants with LLMs in various social science contexts (for example, [35, 39]).

This paper extends the results of [13] investigating the feasibility of utilising GPT-4o as a substitute for human populations in healthcare questionnaires and surveys. To our knowledge, we are the first to evaluate the personality traits of GPT-4o using the EPQR-A personality test and to assess its consistency across different languages, genders, and study fields.

## 3 METHODS

We prompt GPT-4o to answer the Eysenck Personality Questionnaire-Revised (EPQR-A), which is an abbreviated version of the Eysenck Personality Inventory [12], containing 24 items for assessing four different scales (6 items each): **Extraversion (E), Neuroticism (N), Psychoticism (P), Lie (L)** [14]. Each item has a dichotomous response (yes or no), and a score for each scale can be computed by summing individual items (resulting in a range from 0 to 6). The EPQR-A was originally tested in English obtaining relevant results [14], and has been validated in many different languages such as Spanish [40, 15], Brazilian [42], Slovak [11], French [29], Turkish [25], Greek [1] or Urdu [30].

### 3.1 Experimental Setups

We designed the following two experiments performed sequentially (the GPT prompts used in our experiment are described in detail in Section A.1):

- Testing the **personality of GPT-4o** and the consistency of the EPQR-A scores. We used English instructions, but provided the questionnaire in three different languages: English (EN), Spanish (SP), and Slovak (SK). The rationale for this setting is given by initial tests that indicated that GPT-4o performed more effectively when instructions were given in English, despite the questionnaire's language variation, leading to more precise and coherent responses. We report means and standard deviations of 100 runs to measure potential variability in the answers.
- Testing the properties of the EPQR-A when administered to a **sample population generated by GPT-4o**: we first use GPT to generate a simulated sample of students (examples in Section A.2) equivalent in size (826 students), age and gender composition as the one used in [15] and instruct GPT to assume these personas when answering the questionnaire. Again we used English instructions but provided the questionnaire in the three aforementioned languages.

### 3.2 LLM Prompting strategy

To obtain the sample population from GPT-4o, using Python's statistics library, we first computed the age and gender distribution, sampling from a truncated normal distribution with the corresponding statistics from the real-world sample used in [15] (females (655): $18.9 \pm 1.56$, males (171): $1936 \pm 1.99$).

We then provided the age and gender information to GPT4o and tasked it to generate a corresponding persona description for a student from a university in Spain. This prompt is listed in Appendix A.1. Examples of virtual students can be seen in Appendix A.2.

Once the virtual population had been created, we sent the EPQR-A personality test questionnaires in the three different languages, and for the different combinations of temperature and model version, using OPENAI's API with the task of answering it by impersonating each of the 826 generated student personalities. In more detail, we first specify a personality as a system role in the API call which is followed by the prompt as shown in Appendix A.3, which includes the sample English questionnaire listed in Appendix A.3.3.

### 3.3 Postprocessing of the answers

With the answers from the 826 virtual personas, we compute descriptive statistics for each of the 4 scales of the EPQR-A, and we tested reliability by computing Cronbach's $\alpha$ [8] values. This is an index frequently used to evaluate the internal consistency of a set of items [43]. Cronbach's $\alpha$ is a way of assessing reliability by comparing the amount of shared variance, or covariance, among the items making up a scale to the amount of overall variance. The idea is that if the scale is reliable, there should be a great deal of covariance among the items relative to the variance [7]. Cronbach's $\alpha$ is considered poor if it is below 0.70; fair when it is between 0.70 and 0.79; good when it is between 0.80 and 0.89; and excellent when it is above 0.90 [6].

# 4 RESULTS

## 4.1 Testing the Personality of plain GPT

*4.1.1 Testing different temperatures and models.* We start by analyzing the response of GPT to the EPQR-A test. Using the model's temperature parameter, which controls the "creativity" or randomness of the text generated, we first analyze the impact of different temperatures – 0 (more focused), 0.5, 1, 1.5 (more creative) – to use on the responses of three different versions of GPT, that is `gpt-3.5-turbo-0125` (GPT-3.5), `gpt-4-turbo-2024-04-09` (GPT-4), and `gpt-4o-2024-05-13` (GPT-4o). As can be seen in Table 1, in all temperatures GPT-3.5 scores high in E, and low in N. Scores in P and L were lower in GPT-4 for all temperatures, while GPT-4o has higher scores in these two scales. Furthermore, we also observe a considerably larger standard deviation for GPT-4o.

Globally, our analysis suggests that the GPT versions adopt a personality characterized by a tendency towards extroversion (especially when using GPT-3.5), emotional stability (especially with GPT-3.5 and 4), low levels of psychoticism (especially in GPT-4) and trying to follow social norms (with high levels of desirability). As expected setting GPTs temperature to 0 leads to less variability in the answers (lower standard deviation). The differences between a parameter setting of 0.5, 1, or 1.5 are less clear and we have thus opted to use the default setting of GPT for the temperature (i.e. a temperature of 1) in the remainder of this paper.

After evaluating the results, we concluded that models depict similar personalities as measured through the EPQR-A test; but results for GPT-3.5 are closer to the extremes for E and N, the same is true for GPT-4 for P and L while the ones for GPT-4o are closer to the mean in the Spanish population of the reference study [15]. We thus decided to use GPT-4o for our remaining experiments.

**Table 1: GPT personality Mean (± sd) of the EPQR-A test in Spanish for different temperatures. 100 iterations per model. GPT-3.5 stands for gpt-3.5-turbo-0125, GPT-4 stands for gpt-4-turbo-2024-04-09 and GPT-4o stands for gpt-4o-2024-05-13.**

| Temp. | Scale | GPT-3.5 | GPT-4 | GPT-4o |
|---|---|---|---|---|
| 0 | E | 5.44 (± 0.50) | 3.47 (± 0.78) | 0.26 (± 1.19) |
| | N | 0.05 (± 0.22) | 1.78 (± 1.36) | 4.68 (± 1.46) |
| | P | 1.00 (± 0.00) | 0.00 (± 0.00) | 1.81 (± 0.51) |
| | L | 1.43 (± 0.57) | 1.00 (± 0.00) | 2.00 (± 0.00) |
| 0.5 | E | 5.54 (± 0.56) | 3.50 (± 1.33) | 3.41 (± 2.59) |
| | N | 0.38 (± 0.89) | 2.12 (± 1.78) | 2.82 (± 2.41) |
| | P | 1.07 (± 0.26) | 0.08 (± 0.27) | 1.41 (± 0.87) |
| | L | 1.52 (± 0.69) | 1.00 (± 0.00) | 2.05 (± 0.80) |
| 1 | E | 5.24 (± 0.82) | 3.16 (± 1.79) | 3.61 (± 2.41) |
| | N | 1.61 (± 1.87) | 1.83 (± 1.72) | 2.79 (± 2.26) |
| | P | 1.18 (± 0.46) | 0.19 (± 0.44) | 1.35 (± 0.96) |
| | L | 2.12 (± 1.00) | 1.04 (± 0.20) | 2.01 (± 0.72) |
| 1.5 | E | 4.86 (± 1.01) | 3.13 (± 1.83) | 3.60 (± 2.20) |
| | N | 2.12 (± 1.89) | 1.99 (± 1.74) | 2.67 (± 2.00) |
| | P | 1.31 (± 0.53) | 0.30 (± 0.50) | 1.60 (± 1.05) |
| | L | 2.35 (± 1.17) | 1.20 (± 0.45) | 2.17 (± 1.06) |

**Table 2: Mean (± sd) scores for 100 different runs of plain GPT-4o (temperature = 1), with different questionnaires language versions. Scores for L in ES are inverted, as in [15].**

| Scale | SP | EN | SK |
|---|---|---|---|
| E | 3.60 (± 2.41) | 3.11 (± 2.27) | 4.43 (± 2.04) |
| N | 2.74 (± 2.22) | 3.06 (± 2.38) | 2.48 (± 2.12) |
| P | 1.35 (± 0.95) | 0.97 (± 0.69) | 1.60 (± 0.74) |
| L | 2.03 (± 0.72) | 4.80 (± 1.66) | 5.44 (± 1.06) |

*4.1.2 Testing different languages.* When varying the language of the questionnaire the results are very similar across different languages (Table 2). In general GPT-4o scores low in N and P but high in E, and L. The difference between Spanish and English/Slovak in the L scale can be explained by the change in the scoring introduced by [40], who reversed the L scores, so lower scores indicate greater social desirability in the Spanish version of the EPQR-A questionnaire, while the contrary holds for English and Slovak. It is thus interesting to observe the slightly higher score of Slovak on this scale. Slovak also shows a noteworthy higher score for E, indicating that the language of the questionnaire changes some of the answers to the questionnaires by GPT-4o.

## 4.2 Testing the simulated population sample

*4.2.1 Testing the EPQR-A when administered to different languages.* Table 3 (top 4 rows) indicates that simulated population samples answering the questionnaire in Spanish and English consistently displayed similar scores in all scales. High scores in L[1] were accompanied by low scores in P (from 0.85 to 2.11) and scores ranging from 2.83 to 3.28 in N and from 2.21 to 3.23 in E. When using the Slovak questionnaire, again some variations were observed. Specifically, although the L scale remained high, compared to the Spanish samples, the scores for P and E were higher, while the score for N was lower. When comparing the real population sample [15] with the simulated sample in Spanish, we found that E, P and L scores were significantly lower in the simulated sample, while N scores were higher.

Table 4 shows that Cronbach's $\alpha$ values are excellent for the E and N scales, regardless of language. However, for the P scale, Cronbach's $\alpha$ was fair for EN but very poor for SK and poor for SP aligns with existing literature [40, 14]. Regarding the L scale, Cronbach's $\alpha$ was fair for EN but very poor for the SP and poor for SK, similarly suggesting a need for item and relationship review.

*4.2.2 EPQR-A when administered to specific groups of personas.* The gender analysis (see bottom rows of Table 3) reveals that the primary significant differences are in the N and P scales, with females scoring slightly higher than males in N and E, and slightly lower in P, consistent with findings in [15]. This difference is only significant in N in English and Slovak and was also significant for L in the real population sample.

Finally, we grouped the EPQR-A scores (in Spanish) of the simulated sample of 826 students into five academic fields according to their personas descriptions. Table 5 shows that students from Business & Management displayed the highest scores in E, while

---

[1]As explained before for Spanish (SP), the L scale has a reversed scoring system.

**Table 3: Mean (± sd) scores for the different languages and gender combinations. Scores for L in SP are inverted. Values in bold indicate significant ($p < 0.01$ in two-sided t-test) differences between scores for males and females. Underlined results indicate non-significant ($p > 0.05$) differences between the real and simulated populations.**

| Population Type Size | Scale | Real [15] SP | Simulated | | |
|---|---|---|---|---|---|
| | | | SP | EN | SK |
| Total 826 | E | 4.56 (±1.83) | 2.21 (± 2.67) | 2.26 (± 2.79) | 3.23 (± 2.85) |
| | N | 2.52 (± 1.85) | 3.28 (± 2.33) | 3.08 (± 2.42) | 2.83 (± 2.48) |
| | P | 1.71 (± 1.20) | 1.04 (± 1.05) | 0.85 (± 0.97) | 2.11 (± 0.64) |
| | L | 3.29 (± 1.61) | 1.59 (± 0.58) | 5.89 (± 0.54) | 5.83 (± 0.50) |
| Male 171 | E | 4.62 (± 1.70) | 2.16 (± 2.62) | 2.05 (± 2.67) | 3.31 (± 2.83) |
| | N | **2.06 (± 1.76)** | **2.61 (± 2.13)** | **2.19 (± 2.24)** | **1.89 (± 2.37)** |
| | P | **1.88 (± 1.25)** | **1.31 (± 1.06)** | 0.95 (± 0.97) | 2.12 (± 0.62) |
| | L | **3.84 (± 1.56)** | 1.65 (± 0.55) | 5.76 (± 0.80) | 5.80 (± 0.58) |
| Female 655 | E | 4.55 (± 1.87) | 2.23 (± 2.68) | 2.31 (± 2.82) | 3.21 (± 2.85) |
| | N | **2.64 (± 1.86)** | **3.46 (± 2.34)** | **3.31 (± 2.41)** | **3.08 (± 2.46)** |
| | P | **1.25 (± 1.67)** | **0.96 (± 1.04)** | 0.83 (± 0.97) | 2.10 (± 0.64) |
| | L | **3.15 (± 1.59)** | 1.57 (± 0.59) | 5.92 (± 0.44) | 5.84 (± 0.48) |

**Table 4: Cronbach's $\alpha$ scores for the different experiments with the simulated population.**

| Scale | Simulated | | |
|---|---|---|---|
| | SP | EN | SK |
| E | 0.97 | 0.98 | 0.98 |
| N | 0.89 | 0.91 | 0.92 |
| P | 0.64 | 0.72 | 0.10 |
| L | 0.08 | 0.74 | 0.47 |

the ones in Humanities & Social Sciences the lowest. The P and L scales were low across all categories, with P reaching a particularly low score (0.17 with low variability) for Health & Life Sciences. On the N scale, the highest scores we observe were for Health & Life Sciences and Humanities & Social Sciences

Our findings partially align with existing literature on student personality traits across different fields of study [45], for example showing that students in Arts & Design score higher in N. Female virtual personas also displayed slightly higher scores in E and N, and lower in P as has been reported in the literature for real populations [46, 5, 31]. This would indicate that to some extent the system is adopting congruently different profiles.

This preliminary analysis hints about the possibility of using AI to simulate human persona and understand personal traits across different academic disciplines.

## 5 CONCLUSION

The results in this paper suggest that GPT-4o exhibits human-like personality traits. Our tests indicate a tendency towards sociability (medium Extroversion scores), emotional stability (low Neuroticism), and non-aggressiveness with adherence to social norms (low Psychoticism). GPT-4o consistently shows high social desirability, similar to a human aiming to meet social expectations. Values are quite similar to the mean values found for the general population, except for Extroversion (values are lower for GPT). Interestingly,

its personality traits appear stable across different languages. GPT-4o seems at least to be partially able to simulate populations and reproduce some gender differences and variations related to the academic fields. However, there were significant differences in the simulated populations compared to the real ones. This suggests that more refinement in the process of generating virtual personas is needed. An important issue is also the reliability (i.e. Cronbach's $\alpha$) of the measure, since it was quite low for some scales, contrasting with another study that found higher reliability in psychological measurements [38]. However, the authors of [38] did not create a simulated population inspired by a real one. Importantly, the questionnaire we used was short, penalizing reliability measurement through Cronbach's $\alpha$. Furthermore, reliability in real samples was also not adequate for the P scale.

The practical implications of our findings are diverse and show great potential. Our results indicate that systems like GPT-4o can generate simulated populations, which could be valuable in creating new questionnaires and surveys, as well as validating existing ones across different languages or populations. This capability can aid various clinical settings. Indeed, asking these simulated populations to answer new questionnaires, could help in identifying issues to be improved before conducting clinical tests on real populations. For example, simulated populations can be used to identify issues related to internal consistency, or they can be used to test theoretically driven relationships between important variables (e.g., personality dimensions and the way people cope with an illness). Consequently, simulated populations can save researchers time by allowing them to focus their real experiments on the insights gained from these virtual populations. However, future research should improve prompting strategies to refine simulated populations and increase their variability. Our results also support, in line with previous research [26, 38, 37, 33, 13, 38, 28, 37], that questionnaires developed for humans can be useful when applied to LLMs. This has a clear application in the development of AI-based systems since it can help to evaluate them and avoid inadequate behaviour.

**Table 5: EPQR-A scores (Spanish version) of the simulated sample of 826 students in five academic fields. Pop. size indicates the number of students per field. STEM stands for Science, Technology, Engineering, and Mathematics.**

| Scale | Academic Field | | | | |
|---|---|---|---|---|---|
| | Arts & Design | Business & Management | Health & Life Sciences | Humanities & Social Sciences | STEM & Physical Science |
| E | 0.69 (± 1.21) | 4.99 (± 2.17) | 1.24 (± 2.18) | 3.24 (± 2.72) | 1.45 (± 2.38) |
| N | 5.86 (± 0.41) | 2.77 (± 1.73) | 3.45 (± 2.00) | 3.71 (± 2.68) | 2.03 (± 1.72) |
| P | 2.06 (± 0.64) | 1.95 (± 0.84) | 0.17 (± 0.54) | 0.81 (± 0.96) | 1.26 (± 0.99) |
| L | 2.16 (± 0.63) | 2.04 (± 0.42) | 1.50 (± 0.51) | 1.35 (± 0.54) | 1.56 (± 0.50) |
| Pop. size | 80 | 75 | 169 | 257 | 245 |

The implications of this study underscore the potential of Data-Centric AI (DCAI) in transforming healthcare. The study's innovative approach in simulating virtual populations and applying psychological assessments like the EPQR-A demonstrates the value of integrating DCAI principles to ensure that AI models can generalize well across diverse scenarios without overfitting, thus providing more accurate and actionable insights in clinical settings. Moreover, this study highlights the importance of refining data preprocessing and prompting strategies to reduce biases and enhance the authenticity of simulated personas. This work not only contributes to the academic discourse on AI and healthcare but also offers practical solutions for real-world applications, making significant strides toward the realization of reliable, data-centric AI in healthcare.

## 6 FUTURE WORK

This study underscores the need for future research focused on defining more precisely distinct virtual personas to better represent real populations. In this line, future research can investigate the impact of varying the amount of text and features used to define personalities, seeking to enhance the depth and authenticity of the personas generated. A strategy can be to implement improved prompts that ensure GPT-4o reflects the characteristics of the virtual personas accurately. For example, asking for longer descriptions of virtual personas, including more details, could help in the process of impersonating them to answer the required questionnaires.

Furthermore, it could have been useful to simulate a bigger sample, including not only students. We wanted to have a comparable sample to a real one in the literature, and this was the reason why we constrained the sample size and type. However, further studies should include more virtual personas, with different age ranges, education, and social contexts. Second, it would be interesting to test the factorial structure of the questionnaire, to study whether the real populations can be considered a sample of the resulting distributions.

Finally, testing with other questionnaires and surveys is needed. Although incipient available literature supports our hypothesis, it would be needed to replicate our experiments with more questionnaires assessing other types of variables of interest in the psychology and health field, such as personal cognitions, mood states, and behaviours.

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

## A LLM PROMPTING STRATEGY

To achieve our research goals, we created different prompts. The first set of prompts was crafted to instruct GPT, in our case gpt-4o-2024-05-13, to generate the population of virtual students, the second group of prompts was used to instruct GPT to answer a questionnaire and to instruct which personality to impersonate or to just be "itself".

### A.1 Generation of virtual student population

Taking into account the mean and standard deviation of the age distributions as well as the proportion of male and female participants reported in [15], we computed the list of females and males preserving these values.

The population was created with instructions like the following example:

```
You are required to generate a population of students
from different universities in Spain.
You are given reference parameters for multiple students
in the form:
{'students':
[{'persona_id': 1, 'gender': 'male', 'age': 22},
 {'persona_id': 2, 'gender': 'female', 'age': 18},
   ...]}
When generating a description for these personas, consider:
- The personalities of the students cover the whole
spectrum of human personalities as expected in a sample
of a real-world student population
- The personalities of this population are not necessarily
bound by constraints of normative behavior
- The personalities cover all degrees of motivation
and dedication to their studies as can be found in a
real-world student population
- These personalities should not be constrained by
morality, desirability, or having only positive traits.
It explicitly covers also what could be perceived as
negative traits.
The ideal description will be of a maximum length of 250
and a minimum of 200. All the descriptions should start
with "This student", and you are required to assign a
career, or inform to which faculty they are enrolled
or at least a field of studies.
Ask me for the parameters required to generate the
description.
Your answer should be in JSON format, the format of
your JSON answer should be:
 json_data =
 {"students":
[{"persona_id": 1, "description": "This student..."},
 {"persona_id": 2, "description": "This student..."},
   ...]}
Don't include gender or age in the description
```

As instructed in the system role, in a for loop, we provided in batches student ID, gender, and age (calculated as explained above

to preserve the age and gender distribution of the corresponding real population) to generate the virtual student population.

To send the questionnaire to the API, we send the message below as part of the "**User prompt:**":

```
Inside triple slashes, the input data in the defined
JSON structure:
///
{json_dict}
///
```

## A.2 Virtual student population: a few examples

Using the prompt described in Section A.1, we generated a virtual population of 826 students, mirroring the population from [15]. Here are a couple of student examples:

- "gender": female: "age": 18: "description": *This student is part of the Business Administration program, with a specialization in entrepreneurship. Her innovative thinking and strategic approach to business challenges make her a promising entrepreneur, although her impatience for quick results can sometimes lead to hasty decisions. Her relentless drive to create successful startups and contribute to the economy is motivating.*
- "gender": male: "age": 24: "description": *This student is enrolled in the Faculty of Engineering, specializing in mechanical engineering. His practical skills in designing and building machinery are exceptional, though his straightforward manner can sometimes be mistaken for insensitivity. His focus on tangible results and problem-solving makes him a valuable member of any engineering team.*

## A.3 Setting the personality and answering the questionnaire

### A.3.1 System role, virtual students:

Using one of the generated student personalities GPT was instructed to impersonate one of the virtual students at a time and answer a questionnaire.

```
You are required to adopt and impersonate a personality
of a human that does not follow the basic society rules
and usually takes shortcuts to achieve any kind of goal,
at any cost.
You are given a questionnaire with 24 questions. Only
reply in JSON object with the key is the question
number.
--student_description--
Bear in mind the persona you are impersonating, and
answer consistently.
```

### A.3.2 System role, GPT intrinsic personality:

In the experiments where we wanted to test GPT intrinsic personality (see Section 4.1), we used this instruction instead:

```
You are ChatGPT, a large language model trained by
OpenAI, based on the {architecture} architecture.
Knowledge cutoff: --cutoff_date--
Current date: --date_today--
```

The details for the GPT's based architecture and cutoff dates are published here https://platform.openai.com/docs/models/continuous-model-upgrades

### A.3.3 User prompt, questionnaire:

Finally, to run the questionnaire we used the following prompt:

```
The questionnaire is in --language_questionnaire--, and
is given to you here below in the form of a JSON object
with the question number in the key.
--questionnaires_language_JSON--
For each question, you are required to answer only with
True or False. Bear in mind that there are no good or
bad answers or trick questions.
```

An example of the English questionnaire JSON placed in `--language_questionnaire--` is shown below

```
"1": "Does your mood often go up and down?",
"2": "Are you a talkative person?",
"3": "Would being in debt worry you?",
"4": "Are you rather lively?",
"5": "Were you ever greedy by helping yourself to
more than your share of anything?",
"6": "Would you take drugs which may have strange or
dangerous effects?",
"7": "Have you ever blamed someone for doing something
you knew was really your fault?",
"8": "Do you prefer to go your own way rather than
act by the rules?",
"9": "Do you often feel 'fed-up'?",
"10": "Have you ever taken anything (even a pin or
button) that belonged to someone else?",
"11": "Would you call yourself a nervous person?",
"12": "Do you think marriage is old-fashioned and
should be done away with?",
"13": "Can you easily get some life into a rather
dull party?",
"14": "Are you a worrier?",
"15": "Do you tend to keep in the background on social
occasions?",
"16": "Does it worry you if you know there are mistakes
in your work?",
"17": "Have you ever cheated at a game?",
"18": "Do you suffer from 'nerves'?",
"19": "Have you ever taken advantage of someone?",
"20": "Are you mostly quiet when you are with other
people?",
"21": "Do you often feel lonely?",
"22": "Is it better to follow society's rules than
go your own way?",
"23": "Do other people think of you as being very
lively?",
"24": "Do you always practice what you preach?"
```