# OpenReview forum: "How well do simulated populations with GPT-4 align with real ones in clinical trials? The case of the EPQR-A personality test"
_KDD.org/2024/Workshop/AIDSH — KDD-AIDSH 2024 Oral_

### Official Review · Reviewer_te8N · 2024-06-11
**Review for #35**

**Rating:** 5
**Confidence:** 3

**Review:**

### Summary

This paper explores the potential of using GPT-4o to simulate populations for clinical trials, specifically through the use of the Eysenck Personality Questionnaire-Revised (EPQR-A). The authors conducted experiments in three languages (Spanish, English, and Slovak) and analyzed the results to determine the alignment between simulated and real populations.

### Quality

Solid experimental settings and comprehensive results for three languages. The presentation is decent but could be clarified further with illustrative figures.

### Originality and Significance

As a prompt engineering empirical study, the originality lies in the defined problem. There are several works that have explored LLMs' psychology portrayal, e.g.:

- PsychoBench -- On the Humanity of Conversational AI: Evaluating the Psychological Portrayal of LLMs (ICLR 2024 Oral)
- Open Models, Closed Minds? On Agents' Capabilities in Mimicking Human Personalities through Open Large Language Models (arXiv 2024)

There are no apparent differences between these works.

### Writing

Overall, it's clear, but considering the relationship with clinical trials, there is a lack of discussion on how this work's findings address the challenges in clinical trials.

### Pros

- Well-designed experiments on the use of multiple languages with clear objectives.
- Comprehensive analysis with detailed statistical evaluations.
- Latest models are evaluated, e.g., GPT-4o.

### Cons

- Mainly on writing about the impact on clinical trials, as mentioned above.

---

### Official Review · Reviewer_fKTF · 2024-06-12
**Good paper, accept**

**Rating:** 7
**Confidence:** 3

**Review:**

In this paper, the authors performed two experiments on EPQR-A in 3 different languages. The study shows that GPT-4o displays specific personality traits and may vary depending on different parameter settings and questionnaire language.
Pros:
* Systematic experiments: robust design with two distinct experiments.
* Language variations: tested across English, Spanish, and Slovak for broader applicability, and adds depth and originality with multilingual testing.
* Novel application: unique use of GPT-4 in simulating clinical trial populations.

Cons:
* Inconclusive findings: limits practical applicability due to lack of definitive alignment results.
* Lack of details in postprocessing: insufficient explanation in some methodological sections.
* Generalization issues: specific to EPQR-A, limiting generalizability to other assessments.
* Preliminary nature: limits immediate impact on practical applications.
Overall, the paper is well-constructed and contributes to the emerging intersection of AI and psychology. While the results are not conclusive, the study sets a foundation for future research in simulating populations with AI for clinical trials and other applications.

---

### Decision · Program_Chairs · 2024-06-28

Accept (Oral)